# Soil Pollution Characteristics and Microbial Responses in a Vertical Profile with Long-Term Tannery Sludge Contamination in Hebei, China

**DOI:** 10.3390/ijerph16040563

**Published:** 2019-02-15

**Authors:** Xiangke Kong, Chunhui Li, Ping Wang, Guoxin Huang, Zhitao Li, Zhantao Han

**Affiliations:** 1Institute of Hydrogeology & Environmental Geology, Chinese Academy of Geological Sciences, Shijiazhuang 050061, China; kongxiangke@mail.cgs.gov.cn (X.K.); pingwang@mail.cgs.gov.cn (P.W.); 2Hebei and China Geological Survey Key Laboratory of Groundwater Remediation, Institute of Hydrogeology & Environmental Geology, Shijiazhuang 050061, China; 3School of Earth Science and Engineering, North China University of Water Resources and Electric Power, Zhengzhou 450046, China; 1066781547@qq.com; 4Chinese Academy for Environmental Planning, Beijing 100012, China; huanggx@caep.org.cn (G.H.); lizt@caep.org.cn (Z.L.)

**Keywords:** tannery sludge, soil profile, chromium, vertical migration, microbial community

## Abstract

An investigation was made into the effects of tannery sludge on soil chemical properties and microbial communities in a typical soil profile with long-term tannery sludge contamination, North China. The results showed that trivalent chromium (Cr(III)), ammonium, organic nitrogen, salinity and sulfide were the predominant contaminants in tannery sludge. Although the tannery sludge contained high chromium (Cr, 3,0970 mg/kg), the proportion of mobile Cr forms (exchangeable plus carbonate-bound fraction) only accounted for 1.32%. The X-ray diffraction and X-ray photoelectron spectroscopy results further demonstrated that the Cr existed in a stable state of oxides and iron oxides. The alkaline loam soil had a significant retardation effect on the migration of salinity, ammonium, Cr(III) and sulfide, and the accumulation of these contaminants occurred in soils (0–40 cm). A good correlation (R^2^ = 0.959) was observed between total organic carbon (TOC) and Cr(III) in the soil profile, indicating that the dissolved organic matter from sludge leachate promoted the vertical mobility of Cr(III) via forming Cr(III)-organic complexes. The halotolerant bacteria (*Halomonas* and *Tepidimicrobium*) and organic degrading bacteria (*Flavobacteriaceae*, *Tepidimicrobium* and *Balneola*) became the dominant microflora in the soil profile. High contents of salinity, Cr and nitrogen were the main environmental factors affecting the abundance of indigenous microorganisms in soils.

## 1. Introduction

As a dominant traditional manufacture, tannery industry occupies an important part in China’s economy [1]. However, almost one million tons of tannery sludge were generated annually during the tanning process [2,3]. Considering the highly toxic chromium (Cr), the safe disposal of tannery sludge has been becoming an environmental concern in many developing countries [4,5]. Due to the expensive treatment costs and the lack of landfill sites, illegal disposing above the farmland around the tanneries has been a common problem over the past few years in China [6].

With the increasing awareness of environmental protection, local environmental authorities urgently need to treat these tannery sludge disposal sites. Digging out the dumped sludge and its surrounding surface soils seems to be a quick and effective method. However, the deep soils have probably been contaminated by the sludge due to rainfall infiltration and leaching. In addition to highly toxic Cr (containing about 1.0–4.0% of trivalent chromium (Cr(III))) [7], the tannery sludge also contains high contents of salinity, ammonium and organic-nitrogen (organic-N), which have a higher solubility and mobility and further increase the pollution risk to unsaturated zone. Until now, few studies have focused on the existing forms, migration and transformation of these pollutants in vertical profiles, which are important for addressing their threats to the unsaturated and saturated zones [8]. On the other hand, soil microbial communities and activities are sensitive to Cr and nitrogen [9,10]. Considerable studies have used some microbial indicators (biomass and activity) to evaluate the harmful effects of these contaminants on microorganisms in surface soils after application of sludge [11,12,13]. Nevertheless, their effects on the properties of indigenous microorganisms has been neglected, especially for the depth-related changes of microbial community structures in unsaturated zone.

Here, we investigate a typical soil profile (0~200 cm in depth) in a 3-year old tannery sludge disposal site via chemical and microbiological composition analysis. The objectives of this study were to: (1) identify the fundamental properties of the tannery sludge with an assessment of toxic Cr stability; (2) determine the vertical distribution and migration characteristics of the contaminants from the tannery sludge leaching in the soil profile; (3) evaluate the effects of the contaminants on soil microbial properties in the vertical profile.

## 2. Materials and Methods

### 2.1. Field Survey and Sampling

The tannery sludge disposal site was located in a village farmland of Xinji City, Hebei Province, China (37°52′ N 115°15′ E) (Figure 1), which was a major producing area of leather production in North China. The dumped tannery sludge was from neighboring leather industries, which was mainly generated during the tanning process in the past 3 years. There was no artificial seepage control measure in the disposal site. The soil texture in the unsaturated zone was classified as loam according to the international classification standard (Appendix A).

The tannery sludge and the soil samples at the different depths of 0~200 cm in the typical soil profile were taken using soil sampling equipment (Eijkelkamp 01.11.SO). The soil samples for physicochemical characterization analyses were collected and stored in sealed bags at 4 °C. The soil samples for high-throughput sequencing analysis were preserved in pre-sterilized aluminum containers and stored at −70 °C in the laboratory. The uncontaminated soils at the depth of 0~20 cm (US1) and 120~150 cm (US2) were also collected in an uncontaminated site nearby as control (Figure 1). The physicochemical characteristics of the tannery sludge and the soil samples (S1–S8) at the different depths were presented in Table 1. The leachate from the tannery sludge was collected and filtered immediately through a 0.45-μm membrane filter prior to the subsequent measurement.

### 2.2. Chemical Analysis of Tannery Sludge and Soils

All samples were air-dried, crushed, homogenized, and passed through a 0.85 mm (20 mesh) pore diameter sieve before subsequent testing. pH of soil was measured at a soil:water ratio of 1:5 after a contact period of 2 h in an end-over-end shaker (ISO 10390:2005(E)). Moisture of soil was determined using the convection oven drying method. Salinity of soil was determined by measuring the conductivity of the extracts at a soil to water ratio of 1:5 [14]. The total organic carbon (TOC) of soil was determined by the classical potassium dichromate method [15].

The chemical parameters of all samples were analyzed using standard procedures. Total Cr and Fe were determined by inductively coupled plasma atomic emission spectrometry (Thermo scientific iCAP6300, USA) after microwave digestion with a mixture of HF and HNO_3_ solution (USEPA 6010C). Cr(VI) was digested using a 0.28 M Na_2_CO_3_/0.5 M NaOH solution and heating at 95 °C for 60 min (USEPA 3060). Tessier sequential extraction procedures were utilized to evaluate five states of Cr metals existing in the solid phase [16]. Inorganic nitrogen was firstly extracted by 1 M potassium chloride solution, and the ammonium and nitrate were further determined at 630 nm and 543 nm by the spectrophotometric methods (ISO/TS 14256-1:2003 (E)). Total nitrogen was determined by a Kjeldahl method (ISO 11261). Organic-N was calculated as total nitrogen minus the sum of nitrate, nitrite, and ammonium [17]. Cr(VI) was determined at 540 nm using the 1,5-diphenylcarbazide spectrophotometric method (ISO 18412-2005). Cr(III) was calculated by subtracting Cr(VI) from the total Cr content. The sulfide was determined at 665 nm using the methylene blue spectrophotometric method (USEPA 9030B).

A Hitachi S-4800 (Japan) scanning electron microscope with an integrated energy dispersive X-ray system (SEM) was used for morphological analysis. X-ray diffraction (XRD) analysis was carried out using a Bruker D8 Advance brochures diffractometer (Germany). X-ray fluorescence (XRF) was performed to identify the mineral components using a Panalytical axios-advance (Netherlands). The element valences and chemical compositions were measured by X-ray photoelectron spectroscopy (XPS, Thermo scientific ESCALAB 250Xi, UK). Component analyses were undertaken using a Jandel Peak fit software package (an Systat Software Inc., USA). A vario EL cube elemental analyzer was utilized for elemental analyses of C, H, N and O. Fourier transform infrared spectroscopy (FTIR) measurements were performed with a Nicolet IS10 spectrometer (USA) to determine the functional groups in the tannery leachate. The concentrations of dissolved organic matter and organic-N in the sludge leachate were determined by a total organic carbon analyzer (Shimadzu, TOC-L CPH, Japan). The distribution of the relative molecular mass of dissolved organic matter was measured using an on-line high pressure size exclusion chromatography with UV (Shimadzu, prominence LC-20A, Japan) and TOC detectors (GE, Sievers 900, USA).

### 2.3. Microbiological Test Analysis

High-throughput sequencing was conducted for analyzing the microbial community compositions and their functions in the soil samples by Shanghai Personal Biotechnology Co., Ltd. (Shanghai, China). DNA extraction, PCR amplification, purification, pooling, and sequencing of a region of the 16S rRNA gene were performed using standard procedures [18]. Total bacterial genomic DNA samples were extracted using the Fast DNA SPIN extraction kits (MP Biomedicals, Santa Ana, CA, USA). PCR amplification of the bacterial 16S rRNA genes V3-V4 region was performed using the forward primer 338F (5′-ACTCCTACGGGAGGCAGCA-3′) and the reverse primer 806R (5′-GGACTACHVGGGTWTCTAAT-3′). PCR amplicons were purified with Agencourt AMPure Beads (Beckman Coulter, Indianapolis, IN) and quantified using the PicoGreen dsDNA Assay Kit (Invitrogen, Carlsbad, CA, USA). After the individual quantification, amplicons were pooled in equal amounts, and pair-end 2 × 300 bp sequencing was performed using the Illlumina MiSeq platform with an MiSeq Reagent Kit v3 at Shanghai Personal Biotechnology Co., Ltd. (Shanghai, China).

Raw data generated from the high-throughput sequencing run were processed and analyzed following the pipelines of Mothur and QIIME [19,20]. The relative abundance (%) of individual taxa within each community was estimated by comparing the number of sequences assigned to a specific taxon with the number of total sequences obtained for that sample. Clustering of soil samples was based on the nonmetric multidimensional scaling (NMDS) analysis and the unweighted pair group with arithmetic mean (UPGMA) cluster analysis methods. The NMDS ordinations were calculated based on a Jaccard dissimilarity matric [21]. The UPGMA employed a sequential clustering algorithm to identify the local topological relationships in order of similarity, and the phylogenetic tree was built in a stepwise manner [22]. The responses of soil microorganisms to the contaminant factors were evaluated using the spearman rank correlation, and the significant differences of statistical tests were estimated at a significance level of *p* < 0.05 [18].

## 3. Results

### 3.1. Characterization of Tannery Sludge

The physicochemical characterization of the tannery sludge is presented in Table 1. The content of total Cr (30,970 mg/kg) in the tannery sludge was extremely high and above the maximum level (1000 mg/kg) for agriculture or landfill disposal, according to China National Standard (GB4284-1984) [1]. Cr(III) accounted for a principal proportion (>99.5%) of the total Cr in the sludge, whereas the content of Cr (IV) was only 170 mg/kg. In addition, the sludge showed other high-content contaminants, including ammonium (16,080 mg/kg), organic-N (16,500 mg/kg), sulfide (4910 mg/kg) and salinity (99,000 mg/kg). The SEM images exhibited that the morphology of the sludge was coarse, poriferous and of lax structure (Figure 2A). The XRD (Figure 2B) and XRF (Figure 2C) analysis results demonstrated that Cr(III) was mainly combined with calcium salt minerals and crystalline iron oxides in the sludge, including Al_3_Fe_5_O_12_, CaFe_3_O_5_, CaFeO_3_, Cr_2_O_3_, Cr_1.3_Fe_0.73_. The XPS spectrum of Cr (Figure 2D) showed that the observed Cr(2p3/2) and Cr(2p1/2) peaks at 577.6 ev and 587.1 ev correspond to Cr_2_O_3_ and Cr(OH)_3_ [23], which indicated that the Cr(III) was the predominant species in tannery sludge. Combined with the Tessier sequential extraction results of Cr species, the exchangeable (153 mg/kg) and carbonate (229 mg/kg) fractions only accounted for 1.32%, the Fe/Mn oxides-bound fraction accounted for 75.29%, and the strong organic-bound and residual fractions both accounted for 23.38%. There is no doubt that the relatively higher mobilization fractions (exchangeable and carbonates-bound) comprised less than 2% of the total Cr in the sludge. Therefore, Cr(III) was mainly present in the stable oxidation state in the sludge, which weakened the risk of the sludge to nature environments.

As mentioned above, total Cr, ammonium, organic-N, sulfide and salinity in the tannery sludge were considerably high. This would not be accepted for disposal, even in landfills for hazardous wastes [24]. Although the toxicity of nitrogen, sulfide and salinity were less than that of Cr, these contaminants had a great pollution risk due to their higher solubility and mobility. Consequently, it is necessary to assess the pollution risk to unsaturated zone from these released contaminates in the tannery sludge.

### 3.2. Effects of the Tannery Sludge on Soil Physicochemical Properties in the Vertical Profile

#### 3.2.1. The Vertical Distribution Characteristics of Contaminates

As shown in Figure 3, the long-term disposing of the tannery sludge on land has caused a significant contamination to the unsaturated zone, especially the shallow soils of 0–60 cm in depth. The loam alkaline soil had a retention effect on the contaminants transport in the site. The concentrations of the different contaminants decreased drastically with the increase in soil depth. The high concentrations of salinity were still found in the deep soils due to the large release of the soluble salt (chloride) from the sludge [25]. The contents of Cr(III) and Cr(VI) decreased to less than 200 mg/kg and 1 mg/kg in soils beyond 40 cm depth (Figure 3), which was significantly affected by adsorption and precipitation reactions [26]. The Tessier sequential extraction results further showed that the Cr released from the sludge was mainly present as Fe/Mn oxide-bound, strong organic-bound fraction and residual fractions. The exchangeable and carbonate-bound fractions were less than 0.5 mg/kg and 1.3 mg/kg, respectively (Table 2). Accordingly, the high-content Cr(III) released from the sludge was mainly accumulated in the shallow soil (0–40 cm depth), which would have a significant risk to crop planting. 

The concentration of ammonium rapidly decreased to 250 mg/kg at the depth of 60 cm (Figure 3) due to the sorption of fine-grained loam soil [7], whereas a high content of organic-N was still observed in the deep soil (60–200 cm depth). The elemental analysis showed that the tannery sludge had a low C:N ratio of 4.53:1 and a high organic-N content, indicating that the abundant organic-N in sludge was originated from the hydrolysis of the collagen proteins during the tanning process [3]. Moreover, the molecular weight distribution of the dissolved organic matter in the sludge leachate was in the range of 66.77–3856.56 DA, suggesting that the low molecular organic-N accounted for a principal proportion, which had a better solubility and mobility in soil [27]. Therefore, the soluble low molecular organic-N was found in the deep unsaturated zone.

#### 3.2.2. The Correlation between Cr(III) and Total Organic Carbon (TOC)

As shown in Figure 4, Cr(III) had a good linear correlation with TOC (R^2^ = 0.959) in the soils at the different depths. Although inorganic Cr(III) existed as the relatively insoluble Cr(OH)_3_(s) precipitates and was difficult to migrate in alkaline soils [26], the Cr(III) exceeding the background value was observed in the soil at the depths beyond 40 cm (Table 1).

The TOC analysis showed that the sludge leachate had a high concentration (590 mg/L) of dissolved organic matter. The FTIR analysis further illustrated that the leachate contained a series of functional groups, including carboxy, hydroxyl and carbonyl (Appendix A). It can be deduced that Cr(III) can form Cr(III)-organic complexes with these dissolved organic ligands in the leachate and thereby enhance their solubility and mobility in the soils. Other researchers have also reached a similar conclusion [3,27]. In addition, although the soil texture was similar to each other at the different depths (Appendix A), the TOC content in the soil profile had an obvious increase due to the infiltration of dissolved organic matter from the tannery leachate. This phenomenon further suggested that C(III) in the deep soils was mainly present as Cr(III)organic complexes. Clearly, the dissolved low molecular organic matter from the sludge leachate was the main driving force of Cr(III) migration in the unsaturated zone.

### 3.3. Effects of Tannery Sludge on the Soil Microbial Communities in the Vertical Profile

#### 3.3.1. Changes in Soil Microbial Abundance and Diversity

The microbial community compositions and relative abundances of taxa in the soils from different depths were presented in Figure 5 (A, phylum level; B, genus level). As shown in Figure 5A, the dominant microfloras in the soil profile were *Proteobacteria* (13.2–66.3%), *Bacteroidetes* (2.8–69.2%), *Firmicutes* (2.0–35.9%), *Actinobacteria* (0.3–14.9%), which was consistent with the previous research results of the bacterial communities in the Cr-contaminated soils [28,29]. The proportion of each dominant bacterial phylum varied significantly with the soil depths, and the remarkable vertical changes in soil microbial community were found at the different soil depths. Moreover, the microbial community diversity increased with the soil depth increasing, implying that the sludge had a marked effect on the microbial community in the shallow soils.

The genus level characterization further demonstrated the variations in bacterial community in the different soils (Figure 5B). In the tannery sludge, *Halomonas* (49.0%) and *Tepidimicrobium* (11.5%) were the most abundant genera. In the surface soils (S1, S2), *Flavobacteriaceae* (5.4–29.6%), *Tepidimicrobium* (5.5–14.1%), *Balneola* (5.2–8.9%) were the dominant genera. Undoubtedly, the dominated microorganisms present in both of the sludge and the surface soils could tolerate high salinities (Figure 3). *Desulfuromonas* (3.3–11.6%), *Thiobacillus* (3.9–16.4%) and *Desulfuromonadales* (1.2–33.7%) became the dominant genera in the relatively lightly contaminated deep soils (S3–S8), which could be used to illuminate the rapid decrease in sulfide in the soil profile.

The relative abundance and diversity of microbial phyla and genus in the soil profile were significantly different from those of the uncontaminated soils, indicating that the sludge had a remarkable effect on the soil microbial communities in the unsaturated zone

#### 3.3.2. Clustering of Soil Samples Based Microbial Communities

The two-dimensional sort map (Figure 6A) and the UPGMA cluster tree (Figure 6B) were used to compare soils of different pollution levels and their bacterial communities. They both showed that the soils with different contamination levels were distinctly separated, which were divided into 4 distinct clusters: (A) tannery sludge; (B) heavily contaminated surface soils (S1–S2, at the depth of 0–40 cm); (C) lightly contaminated deep soils (S3–S8, beyond 40 cm depth); (D) uncontaminated soils. The sample grouping was markedly correlated with the soil contamination level. Combining with the distribution characterization of the specific contaminates and the microbial community compositions in the soil profile (Figure 3 and Figure 5), the soil contamination level was related to the microbial community owing to the different adaptability of microorganisms to contaminants, and the microbial communities in the surface soils (0–40 cm depth) were influenced greatly by the high content of salinity, Cr(III) and organic-N.

#### 3.3.3. Responses of Soil Microorganisms to the Contaminant Factors

As shown in Table 3, the correlation analysis revealed that the contaminants had the different impacts on the abundances of the dominant genera in soils. *Halomonas*, Tissierella and *Tepidimicrobium* had significant positive correlation coefficient (*p* < 0.01 or 0.05) with salinity, Cr, ammonium and organic-N, whereas *Anaerolineaceae* had a negative correlation with those contaminants (*p* < 0.05).

*Halomonas*, *Tissierella* and *Tepidimicrobium* were closely related to the contaminants in tannery sludge, indicating that these microorganisms had a strong adaptability to the soils contaminated by the tannery sludge. *Halomonas* are a kind of chromate-resistant haloalkaliphilic bacteria, which have been found in the effluents of the tannery wastewater [30]. *Tissierella* and *Tepidimicrobium* are usually observed in high organic-containing sewage sludges [31,32]. On the contrary, *Anaerolineaceae* had a negative correlation coefficient with these contaminants. It was supposed that *Anaerolineaceae* were inhibited due to the high contaminant levels in the soils. As the important bacteria having the capacity of breaking down complex organic matter in active sludges, *Flavobacteriaceae* and *Ulvibacter* were positively correlated with the organic-N [33], which played important roles in the degradation of carbohydrates and organic-N [34]. In conclusion, it was a combination of these contaminants (high salinity, Cr and nitrogen) that affected the abundances of indigenous microorganisms in the soils, and the halotolerant bacteria and the organic degrading bacteria having strong environmental tolerance become the dominant species.

## 4. Conclusions

Although the content of Cr (mainly Cr(III)) in the tannery sludge was very high and above the maximum level for agriculture or landfill disposal, above 98% Cr(III) existed in a stable state of oxides and iron oxides, which weakened its release risk to natural environments. The leachate from sludge containing a large amount of dissolved organic matter caused an increase in TOC content in soils, and a linear correlation was observed between TOC and Cr(III) in the vertical profile. The dissolved organic matter from the sludge leachate was the dominant cue controlling the vertical migration of Cr(III) (forming Cr(III)-organic complexes) in the unsaturated zone. The abundances and diversities of microbial communities were significantly affected by the sludge contamination, and the soils at the different contamination levels could be distinctly separated according to the cluster analysis of the soil microbial communities. The halotolerant bacteria (*Halomonas* and *Tepidimicrobium*) and the organic degrading bacteria (*Flavobacteriaceae*, *Tepidimicrobium* and *Balneola*) having strong environmental tolerance became the dominant species in soils.

## Figures and Tables

**Figure 1 ijerph-16-00563-f001:**
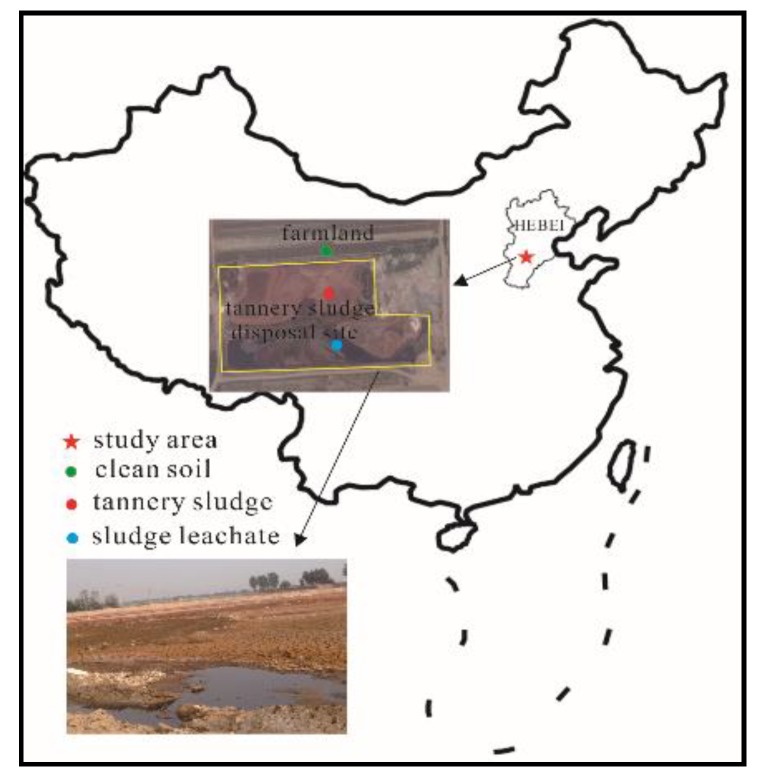
Location of the contaminated site and sites of soil sampling.

**Figure 2 ijerph-16-00563-f002:**
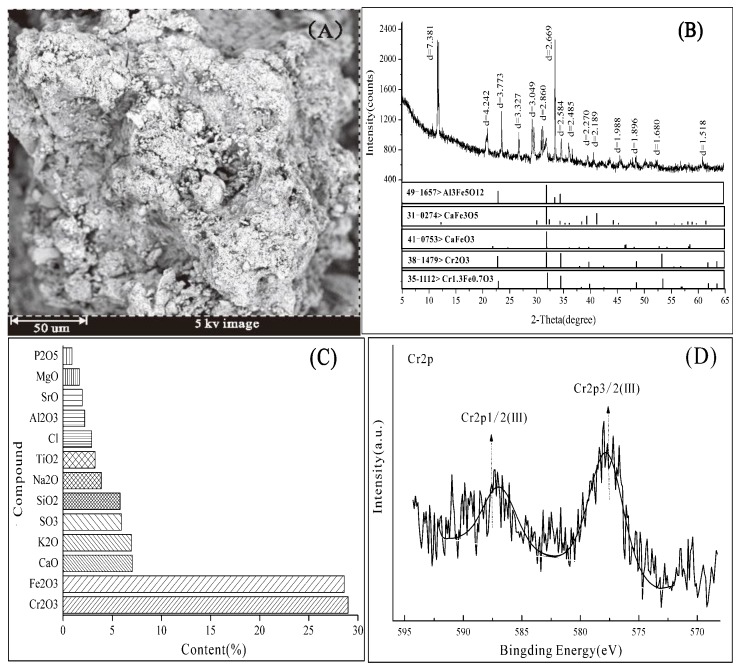
Characterization analysis results of the tannery sludge. (**A**) scanning electron microscope, SEM; (**B**) X-ray diffraction, XRD; (**C**): X-ray fluorescence, XRF; (**D**): X-ray photoelectron spectroscopy, XPS).

**Figure 3 ijerph-16-00563-f003:**
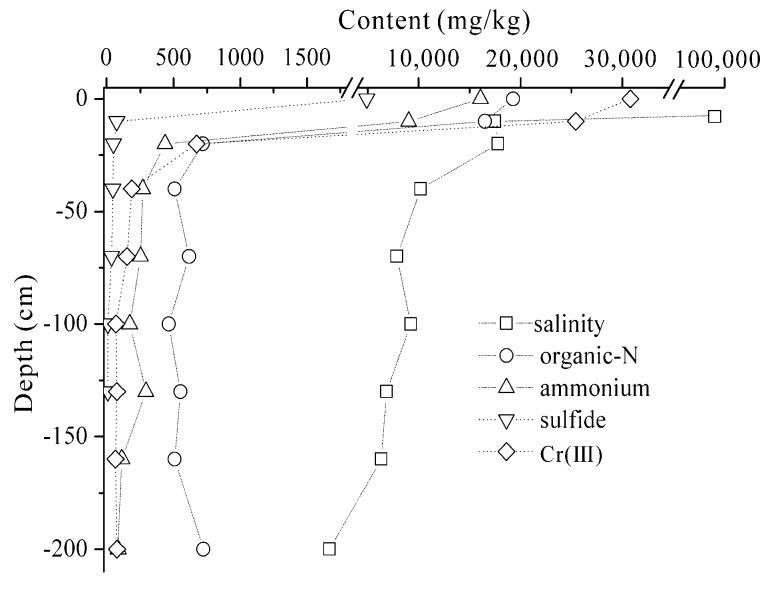
The distribution of major contaminants in the sludge-contaminated soil profile.

**Figure 4 ijerph-16-00563-f004:**
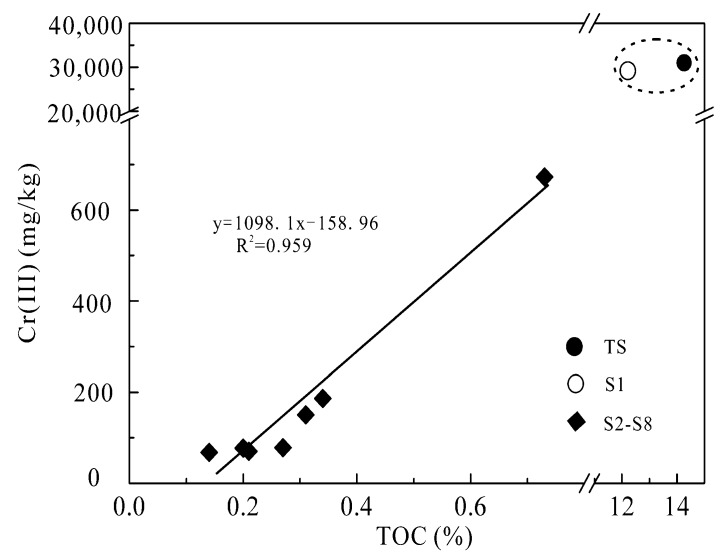
Linear correlation between Cr(III) and total organic carbon (TOC) in the soil profile.

**Figure 5 ijerph-16-00563-f005:**
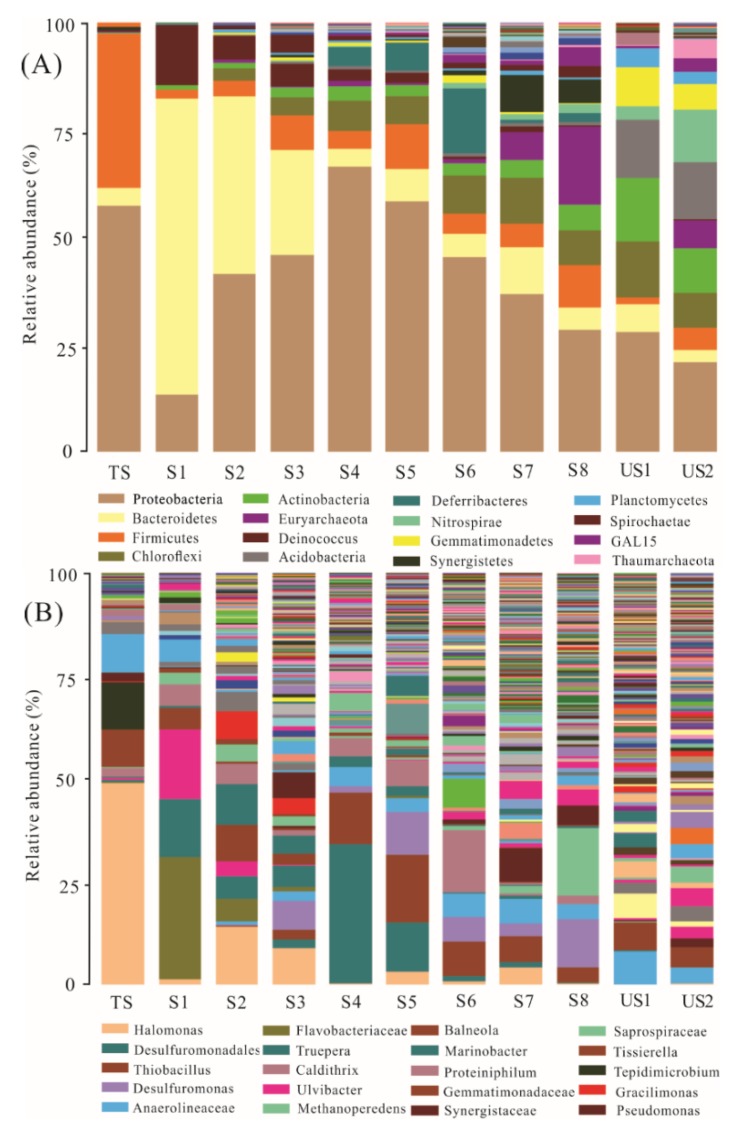
Relative abundances of bacterial taxa recovered from each sample. (**A**) phylum level; (**B**): genus level.

**Figure 6 ijerph-16-00563-f006:**
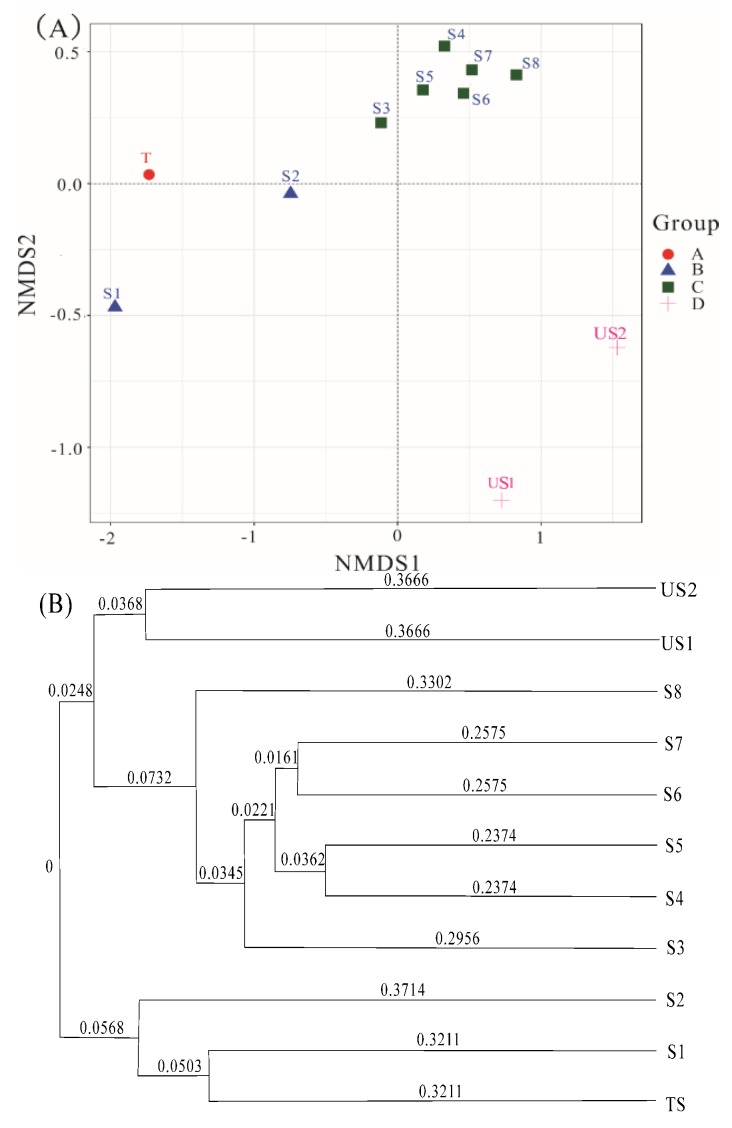
Cluster analysis of soil samples (**A**): the unweighted unifrac analysis (NMDS) plot based on the unweighted UniFrac distances; (**B**) the unweighted pair group with arithmetic mean (UPGMA) plot based on Unweighted pair-group method with arithmetic means.

**Table 1 ijerph-16-00563-t001:** Physicochemical characteristics of tannery sludge and soil samples at the different depths.

Sample	Depth/cm	Moisture/%	pH	Salinity /mg/kg	TOC/wt%	Total Cr/mg/kg	Cr(Ⅲ)/mg/kg	Total Nitrogen/mg/kg	Ammonium/mg/kg	Organic-N/mg/kg	Sulfide/mg/kg
TS ^1^	surface	64.1	7.67	99,000	14.3	30,970	30,800	33,080	16,080	16,500	4910
S1 ^2^	0~20	48.3	7.94	17,500	12.2	25,500	25,427	28,390	9010	19,271	75
S2 ^2^	20~40	17.9	8.37	17,800	0.73	672.61	672	1224	435	719	52
S3 ^2^	40~60	18.1	8.12	10,200	0.34	186.46	184	816	268	507	47
S4 ^2^	60~80	21.1	8.39	7860	0.31	150.61	150	900	252	616	38
S5 ^2^	80~100	18.5	8.31	9220	0.21	70.39	69.8	654	171	463	<10
S6 ^2^	120~150	19.5	8.06	6850	0.20	77.27	76.6	864	292	551	<10
S7 ^2^	150~180	21.5	8.42	6330	0.14	67.76	67.2	630	113	508	<10
S8 ^2^	180~200	18.9	8.20	1670	0.27	78.12	77.5	817	85	723	<10
US1 ^3^	0~20	14.4	8.30	1480	0.23	70	69.86	1210	<20	1178	<10
US2 ^3^	120~150	16.8	8.01	717	0.14	67.2	66.58	154	<20	152	<10

^1^ “TS” means the tannery sludge. ^2^ “S” means a contaminated soil in the vertical profile. ^3^ “US” means an uncontaminated soil.

**Table 2 ijerph-16-00563-t002:** Contents of the different Cr species extracted by a Tessier procedure in the soil samples.

Samples	Depth/cm	Exchangeable/mg/kg	Carbonate/mg/kg	Fe/Mn Oxides /mg/kg	Strong Organic/mg/kg	Residual/mg/kg
Tannery sludge	0	153	229	21,700	3860	2880
S1	10	11.2	124	18,100	7580	3110
S2	20	2.5	6.7	141.6	117.2	353.2
S3	40	<0.5	1.3	40.7	29.4	121.8

**Table 3 ijerph-16-00563-t003:** Spearman rank correlation analysis of the soil contaminants with the dominant genera.

Contaminant	*Halomonas*	*Anaeroli-neaceae*	*Flavobacte-riaceae*	*Ulvibacter*	*Tissierella*	*Tepidimicr-obium*
salinity	0.972 **	−0.579 *	0.003	0.021	0.997 **	0.982 **
TOC	0.667 *	−0.676 *	0.578 *	0.587 *	0.788 **	0.732 *
Cr	0.676 *	−0.667 *	0.563 *	0.572 *	0.796 **	0.743 *
ammonium	0.799 **	−0.643 *	0.388	0.400	0.900**	0.863 **
organic-N	0.524	−0.649 *	0.707 *	0.714 *	0.662 *	0.598 *

** Correlation is significant at the 0.01 level. * Correlation is significant at the 0.05 level.

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
