# Peer review of "Soil Pollution Characteristics and Microbial Responses in a Vertical Profile with Long-Term Tannery Sludge Contamination in Hebei, China"

_ijerph, 2019, doi:10.3390/ijerph16040563_

Round 1
Reviewer 1 Report
The authors present an interesting study of the characteristics of soil pollution and microbial responses after long – term tannery sludge contamination. The work should be on the interest of the readers of the journal. In my opinion, this paper can be published after a miner revision.
General comments:
1. I do not understand the idea of the “Supplementary Materials”. If Authors cite the Table s1 as well as Fig. S1. in the manuscript body, the materials should be immersed to the text in proper place and with the proper numeration consequent to the presented data.
2. I do suggest the proper writing of chemical compound formulas – numbers as subscripts.
3. Although the text is generally understandable, the paper needs slight modification and smooth flow in writing.
Detailed comments:
Chapter 2.1.:
Line 69:
Authors write: “The tannery sludge and the soil samples at the different depths of 0~200 cm in the typical soil 68 profile were taken using a soil sampling equipment (Eijkelkamp 01.11.SO) (Fig.1).” Mentioned “soil sampling equipment” is not presented on Fig.1 like the text suggests
Line 80: Table 1:
TOC is given in [%]. What is that percent?
Header of the table needs some editorial corrections.
Chapter 2.2:
Line 85:
Authors write: “All samples were (…) passed through a 20-mesh sieve before subsequent testing.” I do not understand what was the diameter of sieve, in what units.
Line 86:
Authors write: “ pH of soil was measured at a soil: water ratio of 1: 5”, why such ratio, why not 1:10?
Line 94-95:
Authors write: “Inorganic nitrogen was extracted by potassium chloride solution-spectrophotometric methods”. In my opinion spectroscopic method is the method of nitrogen determination not extraction.
Chapter 3.2.:
In the whole Chapter 3.2. there is not any comparison to another groups of researches results. What causes whole discussion of data no complete.
Line 177: Figure 3:
In my opinion axis of graph should be reversed. If plot presents distribution of contaminants in soil profile the dependence should be content of pollution vs. depth not depth vs. content.
Line 200:
Authors suggest that mentioned organic groups presence is shown on Fig. S1, but groups are not indicated on FTIR graph and readers could be confused. I suggest to indicate wavelengths characteristic of mentioned organic functional groups, because not every reader is a specialist in interpretation of FTIR spectra.
Line 202:
I suggest change “movability” to “mobility”
Line 204:
In the Table S1 there is not indicated the depth, readers have to return to Table 1, what could be confusing. In my opinion Authors of manuscript should add the depth for each profile.
Supplementary Materials:
Table S1.
I do not understand what are, mentioned in the table subject “international classification standard”- USEPA, European Union Directives, WHO or another. Please explain.
References:
Please check correctly the first and second names of reference authors. I am sure that reference [8] authors names are given in wrong way: second author – first name is Zofia, second name Stępniewska; third author: first name - Rafał, second name – Włosek.
My recommendation is checking all references authors.
Author Response
A point-by-point response to the reviewer’s comments is uploaded as a Word file. Please see attachment.

Reviewer 2 Report
Hello,
In my opinion, the work is very interesting from a environmental risk point of view. Cr concentrations are very high! Although Cr(VI) only represents the 0.5%, this value is 170 mg/kg, which is very high and relevant.
However, I miss a statistical methodology section where cluster analysis and Spearman factors should be detailed, not only referenced in the chemical analysis section (line 104). In adittion, methodology used to get results from table S1 related to granulometric distribution has not be mentioned.
Regarding the results, I think XPS results were not discussed in detail.
By last, little things about presentation should be pointed:
In table 1, measuring units were written after "/", but in table 2, parentheses were used...I think the same nomenclature must be used along the manuscript. In adittion, columns width could be modified to improve the appearance of the text, for example, in table 1 Ammonium is written in two lines and could be corrected. This occurs in other tables too.
Line 89, last word: and
Subscripts (lines 92 and 142) are not used and it should be.
Lines 106 and 199: FTIR should be written in capital letters.
Line 202: mobility better than movability.
Lines 204 and 241: It could be better indicates all groups: tannery sludge (TS) and uncontaminated soils (US1-US2)
In general, it is a good work although little things commented above should be improved.
Author Response

(The authors gave the same response as above.)

Reviewer 3 Report
In section 2.2. it says that spectrophotometric analysis were carried out to determine ammonium, nitrate and Cr(VI). Could you add any reference to the methods used?
Table 1 shows values of salinity, organic-N and sulfide, but in section 2.2. the analytical methods of these parameters are not explained. Could you explain how salinity, organic-N and sulfide were analyzed and add some references?
In section 3.1, line 138, the values of ammonium (16500) and organic-N (14400) do not coincide with the values shown in table 1. Review them.
Line 83: You should change “soi” by “soil”.
Line 98: You should change “Cr(III) was calculated by by subtracting” by “Cr(III) was calculated by subtracting”.
In references 2 and 13 the year does not appear in bold.
Author Response

(The authors gave the same response as above.)

Reviewer 4 Report
The work in general is well planned and developed.
If it is pertinent to comment, a deficiency is observed, with the available data they should have linked soil moisture and the effects of tanning sludge on the chemical properties of soil and microbial communities.
On the other hand, the field capacity or saturation humidity of the soil had to be specified.
These observations are based on the fact that water is an important factor in the development of biomass, biodegradation and transport of pollutants.
It should be specified with more quantitative and qualitative background the soil that is worked, beyond the reference of being a free soil, support this with data.
Author Response

(The authors gave the same response as above.)
